# School-Based Digital Innovation Challenges and Way Forward Conversations about Digital Transformation in Education

James Sunney Quaicoe *, Abiodun Afolayan Ogunyemi and Merja Lina Bauters

School of Digital Technologies, Tallinn University, 10120 Tallinn, Estonia; abnogn@tlu.ee (A.A.O.); bauters@tlu.ee (M.L.B.)
* Correspondence: paasanni@tlu.ee

**Abstract:** Background: This article attempts to formulate a school-based model to capitalise on the opportunities and strengths within schools in the Sub-Saharan Africa (SSA) regions to ideate global school-driven digital innovation(s). Consequently, this article explores various digital innovation challenges, opportunities, and elements for schools, as well as proposed school-driven interventions. The paper seeks to open conversations among various international bodies and educational stakeholders, leading to school actors taking ownership of educational projects and school innovation. Methods: A traditional literature review was adopted to analyse the subject of Digital Transformation in Education (DTE). The traditional literature review is a comprehensive and critical overview based on the past and current literature on a subject matter without stringent methodology. Through the literature review methodology, existing materials on the subject matter are subsequently used. Terms and concepts about school innovation and management/leadership were extracted for consideration. These served as a basis for formulating a reference DTE model for interventions. This paper is underpinned by two main conceptual and theoretical bases: (i) The theory of school-based management and its related indicators, and (ii) Michael Fullan's concept of school innovation, which is based on the three key factors of Technology, Pedagogy, and Change knowledge. Fullan's concept is extended to showcase how Active Learning (AL) can inform pedagogical innovation. Results: This paper presents a school-based digital transformation in the education reference model as the outcome. The model uses concept maps to showcase the interrelations between DTE indicators and concepts, and the linkages around which Digital Transformation in Education could be developed as a School-Based Managed (SBM) agenda.

**Keywords:** school-based management; digital innovation school-based innovation; ICTs in education; active learning; digital transformation; active learning; innovative pedagogy

## 1. Introduction

Efforts by governments in Sub-Saharan Africa (SSA) have been significant in an attempt of integrating ICT and education. For instance, in collaboration with GESCI, an international partner in advocating for ICT in Education, numerous activities for digital innovation in schools have been accomplished in SSA Schools. A few cited projects include the Africa Leadership in ICT and Knowledge Society project that ran from 2011–2015. Sixteen Sub-Saharan African Countries participated in the Strengthening of Innovation and Practice in Secondary Education (SIPSE) project [1] that brought ICT integration to schools. In addition, studies conducted in Africa by Evans and Acosta [2] on various educational issues in Kenya, Angola, Uganda, Ethiopia, South Africa, Ghana, Nigeria, and Tanzania ICT in schools showcased the positive roles of technology in the pursuit of education. For instance, in the educational landscape of Kenya, students are provided with e-readers, and teachers and school supervisors are provided with tablets. In South Africa, an innovative approach that offers virtual coaching to teachers is in place, whereas in Zambia, teachers are provided with tablets and schools have digital projectors. Ghana is engaged in broadcasting

live instructions with the focus of transmitting teaching and learning activities to rural areas. Added to these is the deployment of digital tools and resources to schools, and laptops to teachers.

During the COVID-19 health crisis, distinct critical situations emerged between the educational landscapes globally, showcasing disparities existing between rich and developing countries, and in national and regional locations [3–6]. Firstly, the period showcased the digital divide and inequality existing in schools, communities, and regions globally. In Sub-Saharan Africa, a high percentage of schools had to close because they were without any continuing learning or digital learning opportunities (Olaitan et al. [7]). Secondly, that period and its aftermath heightened the interest of education stakeholders in the essence of digital teaching and learning practices in schools. Even though the period was characterised by many setbacks, such as learning losses, school closures, an increase in dropout rates, and pupils/students living in digitally lagged communities being unable to join remote learning sessions, the period ushered in an era for education where the use of digital tools and resources became a national priority for most nations [7,8]. Governments scrambled for recourses to provide online (remote) learning to students, especially in countries where digital teaching and learning had not been ingrained in the educational system. Television, radio, and social media platforms, as well as other innovative approaches, were explored to facilitate learning continuity. Presently, Sub-Saharan countries have turned to digital solutions as a medium to address COVID-19-related issues in the region [9]. The new norm for teaching and learning is defined by digital tools; online, remote, or other digital means [10].

On the other hand, these positive outlooks are not devoid of teething issues. The African continent, as well as some Asian Pacific and Caribbean regions, continue to be saddled with the challenges of digital innovation in education across all educational levels [11]. Pre-COVID-19 educational challenges as observed by researchers continue to persist in post-COVID-19 schools' recovery. Some of the pre-covid era concerns and the associated challenges as observed by researchers are presented in the ensuing discourse. In the works of Willison and Boateng [12], credence is lent to the efforts that stakeholders in ICT education have made in supporting digital teaching and learning schools. However, the findings in this work indicate that digital literacy and technology usage could be higher in schools. Teachers are not using digital tools for teaching and learning activities, and the source of this situation could be how pre-service teachers are trained. At teacher training colleges, not all teachers implement innovative teaching and learning practices in the teaching of their subject courses. Teachers and the Colleges of Education need support, training in digital literacy, and training to integrate ICTs in the teaching of the curriculum in the colleges. The teacher curriculum should provide the professional ICT competence needed by pre-service teacher trainees to use in real-life professional settings, which is technology-driven and characterised by innovative teaching and learning.

Agyei [13] observed that in post-teacher ICT training, teachers mostly used the skills obtained for their personal and professional development rather than in teaching and learning settings. The study infers that transferring the skills and knowledge acquired was met with multiple impediments (including a lack of utilities, digital tools and resources, digital competence, etc.). Based on these discoveries, we believe that it is expedient to involve institutional leaders, actors, and stakeholders in the planning and implementation of innovation and change in organisations. From our personal experiences, we have observed that institutional change and innovation in SSA schools has predominantly been a top-down approach. The implication is that most schools lack the initiative to innovate without looking for "orders from above".

Most SSA governments acknowledge the official integration of ICTs in education. However, as observed by Ngajie and Ngo [14], there is a need for multi-stakeholder sensitization of ICT in schools. Emphasis should be put on involving parents, creating linkages to pedagogy, technology, and curriculum, and creating structures for shared interpretation of ICT policy in schools, as well as clarifying the concept of ICT in schools to

stakeholders. Previous research by Tondeur et al. [15] raised concerns about the negative impact of ICT in education if the following are not adequately addressed: utility provision to enhance the use of digital tools and resources, a vision formulation to direct implementation, and consistency in the interpretation of ICT in education among staff/stakeholders and school leaders.

In line with the preceding discourse, we are constantly asking ourselves "Is school digital innovation in schools all about technology?" Mukuni [16] helps in answering the question. During a post-COVID 19 research, the study concluded that technology in schools did not inform the teacher of digital professional development and learners' literacy, and did not address the digital divide across schools. Meanwhile, other challenges persist in schools that render the technology ineffective. For instance, Evans and Acosta [2] inferred that some schools lacked basic utilities and the internet needed to support technology usage in the schools. Despite numerous opportunities, threats, and challenges, SSA has shown the will to innovate. There is a need for research support in securing the future of digital technology in African schools and in regions where similar trends exist. UNESCO [17] governments and educational institutions continue to provide resources to create a digital learning ecosystem in the region for technology in education. In this light, we seek to open further conversation about the subject matter and pitch the notion that even when there are digital services for schools to access, there is a need for a plan of action and commitment on the part of schools to benefit. This study advocates for a kind of school-driven local governance approach that will support: (i) formulating policies that support school access to and utilisation of the digital infrastructure and resources; (ii) training teachers for technology uptake and professional usage of digital tools; (iii) empowering teachers to be agents of change through innovative pedagogical practices; and (iv) pursuing a digital innovation agenda with customised management approaches that are relevant to the needs of the school and its immediate community. Governments can support and address real needs based on bottom-up inputs if schools are empowered to measure their needs and formulate innovation plans [18].

## 1.1. Problem(s) and Contested Issues

We share a similar position with Gondwe [19], that designing and implementing Policy for ICT integration should be based on research and not speculation. A global study conducted by Conole [20] showed how the policy implementation approach influences the practice and outcome of the intended reform(s). Top-down policies often lead to some sections of society being left behind, as observed by Dube [21]. Such neglect could be avoided; Conole's findings indicate that the drivers of a successful ICT in education entail harmonising policy context, and having policy directives and actual practice plans that afford stakeholder participation. We are of the view that there is a lack of empowerment at the school level for digital innovation and the absence of empowered school actors is preventing most schools from being innovative. Practically, digital transformation in education cannot be overlooked, because digital tools and resources continue to shape all the spheres of educational practices; the teaching profession, student learning and competence development, attainment of learning outcomes, and the learning processes and assessments are all affected. The next and upcoming generation essentially needs digital literacy skills for their survival and for economic liberation [22]. In the work of Hubenakova et al. [23], formulating institutional digital transformation is focused on three areas, namely: (i) developing the digital competence of staff and students, (ii) provision of digital settings or environment teaching–learning activities, and (iii) compilation or provision of a repository of digital experiences. Inferring from our post-COVID-19 experience, the new norm for teaching and learning is defined by digital tools, materials, and processes (online, remote, or other digital means) [10].

ICT in education in Africa cannot continue to be top-down driven; schools need empowerment to ideate and innovate, and the post-COVID-19 educational scramble experience has shown the way. Many countries used locally based approaches to promote

continuity in learning during the pandemic period. In some countries, Public–Private collaborations were solidified to promote remote teaching and learning. Consequently, the objective of this article is to open a conversation that would articulate the various digital innovation challenges and opportunities for schools, and recommend solutions for a base school stakeholders' empowerment. In effect, this current paper seeks to initiate further conversations among stakeholders of education and interested parties associated with digital transformation in schools. The conversation is guided by the following questions:

- What DTE concerns are found in schools and what are the concerns impacting school innovation?
- What could be the possible contents of a DTE framework for school innovation?
- What could be the focus of a typical locally school-driven DTE agenda?
- What DTE broader framework could be proposed to serve as a reference model for school actors to plan their school-based management agenda?

### 1.2. Conversations about Conceptualising Digital Transformation in Education in Schools

The subject of digital teaching and learning innovation comes with ambiguity in definitions. The temptation to relegate the idea of innovation in schools to just technology in teaching and learning is high. Additionally, other terms or phrases such as digital teaching and learning, digital transformation in schools/education, digital innovative teaching practices, computers in education, innovative teaching and learning, and innovative digital teaching and learning also make the definition of this subject of digital actions in school innovation very complex to define. Considering this hazy picture, this current paper compresses the idea of digital teaching and learning, or school digital innovation and its related thoughts, under the umbrella term "Digital Transformation in Education—DTE", with the contextual framework of DTE generated using the following sources. For Reis et al. [24] the term Digital Transformation is defined with three themes, namely: (i) Technological, (ii) Organizational, and (iii) Social. The technological elements are digital infrastructure and resources. The organization-focused elements embrace change, innovation, and/or the introduction of new products or processes or business models. Finally, the socially focused elements are about end users/beneficiaries/actors or human life.

Additionally, Sousa and Rocha [25] researched digital transformation in education and advanced two scenarios associated with DTE as follows: (i) Digital transformation for digital learning, which is about leveraging the affordances of mobile phones, tablets, smartphones, and all smart applications to facilitate learning, and (ii) skills for digital transformation, relating to artificial intelligence, internet of things, virtual/augmented realities, nanotechnology, and robotization as engines for teaching and learning. Furthermore, Patton and Santos [26] researched the actors and the elements of digital transformation in an organisation, concerning the school organisation context. The elements of DTE were identified as the beneficiaries of the transformation in education, the teachers facilitating the education, the curriculum defining the process and contents of the education, and the institutional culture. Insights gained so far from literature give a convincing notion that digital transformation in education is more than just technology. With this backdrop, we hold the view that in defining digital transformation in education, the following should be included: (i) the student, (ii) the teacher, (iii) the curriculum, (iv) the teaching and learning activities and processes, (v) the management of instruction, (vi) safety in pursuit of an educational agenda, (vii) what is effective and what is not; and (iii) the kind of school culture being sought.

From Cambridge Education [27] digital transformation in education as viewed by Innovation in Education (INED) is described as "New ideas and methodologies within the context of the initiative that is being proposed, this might involve completely novel approaches or (proven) ideas and/or approaches taken from other contexts and adapted to the current" [27] (p. 8). The takeaways from these viewpoints of DTE for consideration are (i) the existence of an idea or situation that needs to be enhanced, (ii) a methodology, approach, or process to be improved, and (iii) the deployment of resources (inputs)

and harmonisation of the inputs to improve the situation. Considering the preceding discourse about digital transformation, this paper projects the following themes as areas characterising digital transformation in education (although not an exhaustive list):

- Digital infrastructure, tools, and resources (availability and uptake)
- School as an organisation or a system (interrelated components)
- Social interactions, including teaching and learning activities (social network of school actors)
- Various literacies, competencies, and school actor's pre-defined dispositions/traits
- Learning/instructional management
- Learning environment (convenience and safety)
- Decision makings (research/data-based)
- Context of change (innovation agenda)
- Change facilitation (managing the innovation)

Concerning the pre-defined characteristics, this paper reemphasizes the term Digital Transformation in Education (DTE) as synonymous with school digital innovation or digital teaching and learning innovation or ICT in education, and defines DTE as "Using digital/ICT tools and resources as leverage to operationalise school activities, including (i) planning, developing, administering, and managing educational contents and processes, (ii) innovating learning/instructional design, delivery, and evaluation, (iii) supporting learning continuity and learner progression and achievements, (iv) facilitating institutional efficiency through data-driven (intelligent) decisions, (v) supporting the well-being of both staff and students and (vi) providing a basis for trustworthy settings for actors (both staff and students) and stakeholders to be empowered to co-create and implement educational outcomes".

### 1.3. Digital Transformation in Education (DTE) and the 21st-Century's Pedagogical Skills

In DTE the overarching focus is the provision of 21st-century skills across all subject domains to prepare learners to fit in a knowledge-driven society, and for their overall well-being; pursuing 21st-century skills pedagogy in schools should be a principal focus in school innovation. As a result, schools should deliberately or consciously provide a setting for 21st-century learning to occur; otherwise, innovation will stagnate. Making inferences regarding the general stagnation in digital transformation in educational institutions and business organisations [28] presents interesting insights into the situation. The authors [28] assert that there is (i) a lack of vision and institutional innovation strategy, (ii) an absence of motivation to propel or initiate innovation, (iii) a lack of adequate competence needed for innovation, and finally, (iv) the will for institutions to redefine roles and create learning opportunities and innovations for change. With this backdrop, we hold the view that for an institution to ideate for DTE, school actors and stakeholders must have a shared and common understanding or vision about the connections between digital transformation in education (schools) and the literacies that informs DTE for pedagogical innovation. After all, the success of a school's 21st-century competence readiness is what will allow learners to thrive in a 21st-century world [27].

In Figure 1 the skills or literacies for 21st-century pedagogies are presented. The schema represents an extract from Partners for 21st-Century Skills [29]. This framework offers a basis for various stakeholders in education to have a common understanding of the skills to be integrated into subjects and pedagogical practices. In this case, teachers are offered a referencing point to associate technology with the skills anticipated regarding national curriculum requirements. It has already been established in this discourse that there is an inconsistency in the interpretation of what goes into DTE. Consequently, the educational stakeholder is at loss as to what are the starting point and limits of school digital innovation. Accordingly, a shared vision is essential for any DTE agenda in a school to be successful. As observed by Balyer and Oz, [30], stakeholders in education need to appreciate and accept that the use of digital tools and resources will always push the teaching profession and the field of education to transform. Therefore, for institutional

transformation to occur, there should be a digital innovation vision or strategy. This vision must be shared with all stakeholders in the institution; through this collective responsibility, it is possible to initiate a shared innovation journey in schools. Thus, we assert that in pursuing the DTE agenda, stakeholders in education should be supported, motivated, and empowered to work together.

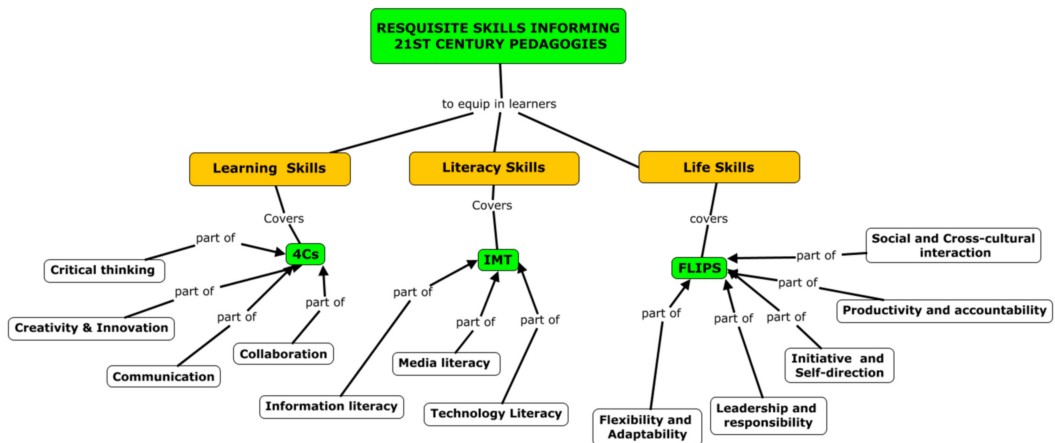

**Figure 1.** Schema informing of 21st-Century Skills and pedagogies, designed by the author with ideas extracted from [29,31,32].

Consequently, the quest for digital transformation in schools would become a transparent and shared vision when conversations are had as a team about DTE. The conversations should centre on 21st-century teaching and learning pedagogies, the digital culture of the school, drivers of digital innovation, digital tools and resources, and new processes that are all needed for the change.

As indicated by Wilson and Boateng [12], pre-service teachers have challenges in practising their profession in the real world of work because the circumstances surrounding teacher training activities tend to be different from the workplace. Inquiry is needed to help re-orient teachers to be effective in their professional practices. For schools to be able to ideate and innovate [30], a vision is needed, and inquiries are good starting points for the creation of the school visions. This paper directs stakeholders to be guided by The Partners for 21st-Century Skills [29] framework. Applying the contents of 21st-century learning and pedagogies, stakeholders at the grassroots level will be able to develop a shared vision and a common understanding of what could go into Digital Transformation in Education in their respective schools.

### 1.4. Asserting the Need for a School-Driven Digital Transformation

We observed that government contributions and support for ICT in schools in the SSA region have been positive. However, the deployment of laptops, desktop computers, and, in some cases, tablets have been over-emphasized at the expense of training the school to innovate using the resources. Therefore, we singled out the following viewpoints as the reason why the conversation about DTE should be continued in the corridors of the Education Ministry, Regional, Municipal and District Education Offices, in schools, and among international agencies operating in SSA.

- The subject of digital innovation in schools to incorporate 21st-century Learning pedagogies is not about speculations; rather it should be pragmatic, based on facts (research), vision, and leadership [33,34]. Therefore, the school needs baseline evidence of the digital disparities that should be addressed, and the strengths and opportunities at the school's disposal.
- Leadership for innovation should be the stakeholders' and school actors' shared vision and responsibility [35]. Most schools lack this collective vision for innovation

and this trend needs to be corrected. The distorted vision of DTE pervades the educational hierarchy.

- 21st-century skills for learners cannot be developed in isolation, i.e., alienated from other subject disciples and factors such as teacher digital literacy, learning resources, learning agenda, and processes, to mention but a few. They should be harmonised, through collaborative dialogues and teamwork [29].

- The need for teacher professionalism that will lead to training students for life lies in creating an enabling development environment that is characterised by opportunities for creativity, critical thinking problem-solving, including all other 21st-century competencies [36]. The teacher needs a stage to measure digital efficacy, and professional readiness to innovate and receive support to intervene on limitations.

- The provision of resources alone does not guarantee reform success; however, good leadership together with resources does [37]. Motivation through participatory leadership breeds shared vision, empowerment, and ownership of the school innovation agenda, culminating in successful DTE.

### 1.5. Fullan's Educational Change Model

Michael Fullan's model of educational change focused on three areas, namely Pedagogy, Technology, and Change Knowledge. The inception of the model was a concern Fullan had about the school's system and how teaching and learning activities were conducted [38]. His concerns were (i) schools and the processes involved not being dynamic but rather regimental (schools organised with strict structures), (ii) education not transmitting relevant content and using wrong approaches, and (iii) learning made less exciting to learners because it is not engaging and motivating [38,39].

In the context of promoting digital innovation in schools, Fullan advocates for learner-centred pedagogies that are interactive and motivating. Teaching and learning activities need to pave a way for strong teacher-student relationships. Thus, to achieve pedagogical innovation, technology should be the driving force that creates affordances and facilitates innovation in teaching and learning. Technology should bring the teacher and the learner together as co-creators in the pedagogical process. The implication is that technology should be the medium for empowering the learner and giving a platform to articulate thoughts and develop competence. Despite the positive role of technology in pedagogical innovation, Fullan was also mindful of the inherent disadvantages that technology brings to teaching and learning. He advocated for the need to guard against the misused or misplaced application of technology in schools. Finally, in the education change model, Fullan considers the change knowledge components as the part of the model where the planning, initiating, and implementation of the reform (innovation) idea occurs. It entails harmonising every component of the education change or innovation for the reform to be a reality [38].

Situating the concerns of Fullan in the context of this article, we claim that due to prevailing challenges in the educational landscape (which tend to be more visible at the schools), it is expedient for governments to support school-based initiated solutions. This call is based on the premise that schools have very specific needs and challenges which global or wholesale interventions of governments fail to address. For instance, in the SSA educational landscape, the literature holds that:

- Teachers have the desire to implement ICTs in their teaching in learning activities but have school-based challenges that appear to inhibit them from doing so [40].
- Schools encounter infrastructure challenges in the quest for innovation [16].
- Access to technology alone cannot improve learning, provided that the needed affordances to meet interventions are not forthcoming [41].
- There are issues surrounding ICT in schools, especially in the areas of policies for integration, teacher competence, teacher confidence, motivation and incentives for integration, teacher beliefs, infrastructure, teacher training for digital pedagogical

skills, availability of utilities, political situations and ICT usage, and curriculum alignment [42]
- Lack of understanding among stakeholders about the meaning of ICTs in schools [43].

In sum, we advocate for conscious efforts on the part of the schools' actors to have an innovation vision (school digital agenda), identify leadership for innovation, obtain resources and expertise, and undertake the innovation journey as a team, in the framework of a school-managed agenda [15,44].

### 1.6. Active Learning (AL) for Pedagogical Innovation in Fullan's Model

As already alluded to, Fullan argued about schools not being dynamic but rather regimental, transmitting irrelevant content and applying wrong approaches to the teaching and learning activities; further, he claimed that learning has been made less exciting to learners because it is not engaging or motivating [38,39]. Therefore, in parallel, we considered that in the search for DTE in schools, a student-centred learning approach is the way to go. These student-centred approaches should be incorporated into the innovation to orchestrate pedagogical innovation.

We propose Active Learning (AL) as one of the ways to address pedagogical concerns raised by Fullan, which was about learning not being engaging enough for the learners. In the work of Chickering and Ehrmann [45], seven principles of good practice in under-graduate education are advanced. We focused on the parts that correlate with Fullan's viewpoint; these are (i) a good learning practice should be engaging, (ii) a good learning practice should offer peer cooperation and collaboration in the learning spaces (thus, learning should be an active activity for the learners), and (iii) good learning practices include giving learners feedback in their learning trajectories. By making AL the foundation for pedagogical innovation, learners' knowledge and understanding of the value and role in the teaching–learning spaces is enhanced. The learner gains a sense of responsibility because the learning is personalised, and some level of learning autonomy is afforded. Beaudry [46] observed that Active Learning is used by educational institutions for various reasons, such as teaching and learning, leadership training, capacity development, and facilitating stakeholder involvement and strategic planning activities. AL leads to increased student learning satisfaction, due to its affordance of interactivity in learning. As already indicated, AL also supports staff development as it makes learning more authentic and experiential.

For Verdiyeva [47], AL is essential in educational practices to the learner because communication and adaptation for the learning enhance the form of personal-oriented learning ownership, which equally appeals to the student as much as the teacher. Regarding the relevance of AL to the teacher, Sitthiworachart et al. [48] observed that teachers' self-awareness and actual experiences about the use of technology to facilitate AL lessons enhanced their comprehension as to why they have migrated from conventional teaching–learning practices to technology-supported innovative pedagogical practices.

To design AL-induced pedagogical innovations, which are intended to address the concern of regimental pedagogies in school, we took a cue from the work of Bonwell and Eison [49] to outline the characteristics that should underpin lesson design:
- Students/learners should be active participants in the learning, not mere listeners of the teacher.
- Students/learners should be allowed to make use of their ideas and skills in practical learning spaces, not simply receivers of information.
- Students/learners should be offered opportunities to engage in higher-order thinking activities, such as analysing, synthesising evaluating, and making facts-based decisions.
- Students/learners should be allowed to engage with both peers and the teacher/facilitator.
- Students/learners should be offered the opportunity to make personal inquiries about their learning responsibilities (metacognition), values, attitudes, previous knowledge and experiences, and dispositions.

*1.7. School-Based Management*

School-based management (SBM) could be described as a systemic interdependency of local school stakeholders working together to interpret and implement the national educational policy at a grassroots or local level. It is worth noting that SBM is not about schools assuming the national/government roles in education delivery or policy implementation; rather, it is about making the national policy implementation relevant to the diverse local and school situations [50]. Thus, according to Malen as cited by [50], "School-based management can be viewed conceptually as a form of alteration of governance structures, as a form of decentralization that identifies the individual school as a primary unit of improvement and relies on the redistribution of decision-making authority as the primary means through which improvement might be simulated and sustained." In sum, SBM is defined in this article as a governance model where authority and decision-making in schools are decentralized, giving control and management of the business of the school to the principals/headteachers, teachers, parents, students, and selected community members [51].

Creating models of school-based management can be complex and inconsistent owning to what kind of interpretations are given to the approach and various conditions surrounding its practice. Accordingly, [50] suggests typologies within which SBM can be observed in the context of leadership are (i) who has the control and the decision-making power of the school, and (ii) the degree of autonomy of the school or within the school. On a continuum, the SBM model stretches from limited autonomy at one end to absolute autonomy at the other. These are synonymous with Weak SBM to Very Strong SBM. In between the strong and weak SBMs are other statuses, namely Moderate SBM, Somewhat Strong SBM, and Strong SBM. Weak SBM stands for no autonomy (total government control); Moderate SBM is characterised by limited autonomy but not in key decision-making powers; Somewhat Strong SBM is characterised by school boards having the autonomy to advise dealings in the school and having some decision-making powers in school management; Strong SBM is marked by councils and schools having controlling powers where the community, parents, and staff make decisions about what happens at the schools; and Very Strong SBM occurs when schools having absolute control in decision-making and operate autonomously.

In the works of Moradi [33], school-based management practices have been identified as correlating with the success of school reforms and are influenced by the presence of a well-defined vision or agenda that gives a clear explanation of which reform is anticipated or targeted. In another study about SBM, Squires and Kranyik [52] asserted that SBM offers working teams the opportunity to build sustained and supportive bonding that generates positive impacts around instructional progress.

Furthermore, some research [53–55] concluded that SBM brings along favourable characteristics relevant to school reforms. For instance, it is observed that SBM has the potential to (i) improve how school governance is practised, (ii) promote stakeholder participation in decision-making for reforms, and (iii) offer the opportunity for national policies to be tailored to fit or address local and unique community circumstances. In addition, SBM has the potential to enhance the actors' commitment to the reform and makes the implementation and management of reforms a bit more flexible.

It is also worth noting that SBM does have a share of opposing factors, and [56] observed that factors such as (i) the scope of the reform being pursued, (ii) the extent to which government/national structures support the reform, (iii) the parental acceptance of the reforms, and (iv) actions by anti-reform unions. Typical examples of a failed SBM agenda as observed from the works of Fullan and Watson [39] include: (i) the reform was implemented but teachers stuck to their traditional and comfortable professional approaches; (ii) the implemented SBM reform did not bring about the expected learning innovation in the classrooms; probably owing to internal resistance or external influences, (iii) conflicts between SBM visions and National/Regional/District Educational management bodies; and finally, (iv) the tendency of SBM losing focus on the core vision for innovation in the

schools as aligned to the national vision. As an alert mechanism, stakeholders need to understand that in the educational system, complete autonomy for SBM is an illusion; some aspects of national/regional/district education oversight will always prevail. Therefore, for the school system to embark on the SBM agenda for digital innovation, there should be a vision which must be shared with all relevant stakeholders. This may then be followed by negotiations and consensus building, and providing checks and balances towards functionality and protocols of the SBM agenda, to allay the fears of the educational managers at the Local and Regional Educational Management levels and the opposing stakeholders.

## 2. Materials and Methods

The research design type adopted in this work is a simple traditional literature review [57]. A traditional literature review is a comprehensive and critical overview based on the past and current literature on a subject matter without a stringent methodology. As a result, the approach supported the gathering of related published materials, undertaking elaborate narration and conversation (report inferences) about the subject matter under review. In the selection of this approach, a way was paved for an analytical review that extracted best practices and models to showcase our position regarding the subject matter of digital transformation in education. Consequently, several papers and reports were gathered from various sources that included the Scopus and Web of Science databases, and in addition selecting other relevant materials using the Google Scholar search engine. Sourced materials had publishers such as Oxford Journals, Springer, Francis and Taylor, and Science Direct, among others. Practically, the extracted texts could be classified into two categories, namely, (i) peer-reviewed materials (including journals, articles, and conference proceedings), and (ii) non-academic materials (not peer-reviewed, including relevant contributions from books, reports, websites, and policy documents from globally recognised educational bodies, among others). It is worth noting that the choice of publications was contingent on the keywords (themes) associated with the subject. As much as we aligned ourselves to scholarly reasoning, we wanted this paper to be more of an unrestricted conversation based on academic extracts (the literature) as well as from other stakeholder sources. Consequently, the search and selection criteria were based on the following themes: school leadership, school management, education, school-based activities, digital teaching, digital innovation in schools, digital transformation, ICT in education, digital teaching and learning in schools in the SSA Region, and ICT in schools in the SSA Region; as well as other regional and global sources. An initial collection of 250 papers were extracted from the mentioned sources (Web of Science, Scopus, and Google's search engines).

The 250 papers were further screened using very specific key works, with emphasis on the words and group of words strongly relative to the objective of the current article; for instance, articles addressing "technology innovation in schools", and/or "school-based management to digital innovation", etc. (see Figure 2). This bought the selected papers/documents/books to a total of ninety-one (91). The 91 selected publications were further clustered into the contexts of how they address the subject matter of the conversation (school innovation issues from technology and pedagogical perspectives, and how they portray school-based management of digital innovations). The 91 selected publications (both peer-reviewed and non-academic) were subjected to further content analysis to ensure that the papers extracted offer relevant inputs to the questions guiding this current article. Figure 2 offers an illustration of the process followed to extract the DTE focal themes, which are (i) pedagogically focused innovation, (ii) school-based leadership for innovation and change management, and (iii) digital technologies for innovation and change facilitation.

In the final analysis, concept maps were used to showcase the extracts made from the literature review, which is meant to be a reference model for DTE conversation among school stakeholders. With support from the literature, the insights gained constituted the results of the exploration. It is worth mentioning that all 91 publications supported the themes, but not all supported in-depth contributions to the building of the DTE model. Therefore, those whose bulk inputs constituted the subject matter were referenced accordingly.

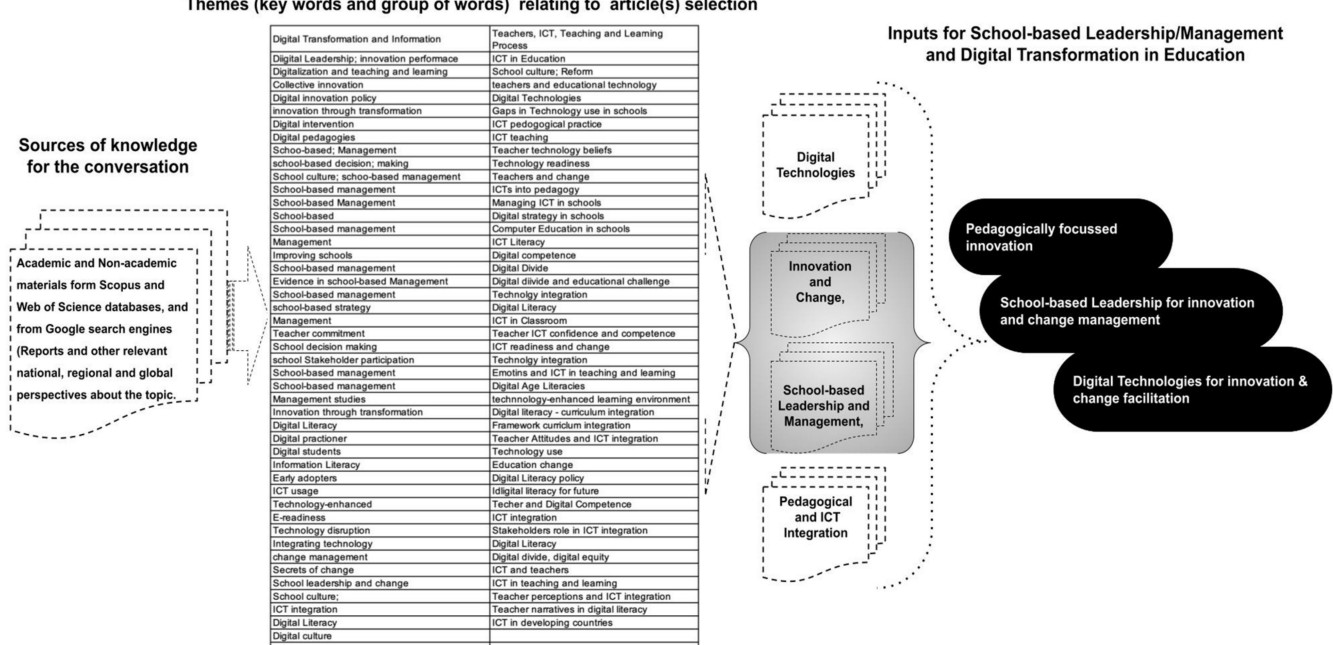

**Figure 2.** Schematic representation of the methodology used in the article to extract school-based Digital Transformation Factors.

## 3. Results

In the context of the reviewed papers, we present the results as the position of the authors to ideate for school-based digital transformation. This paper is underpinned by two main conceptual and theoretical bases: (i) Michael Fullan's concept of School innovation based on three key factors of Technology, Pedagogy, and Change Knowledge [39,58,59]. Fullan's innovative pedagogy is extended with AL from the works of Bonwell and Eison, as well as Chickering and Ehrmaan's [45,49] Active Learning thought. (ii) The concept of school-based management and its related indicator elements, actors, and processes for the successful accomplishment of activities [50].

### 3.1. School-Based Digital Transformation in Education

In Figure 3, the schema for the school-based digital transformation model patterned along the thought of school innovation and change in the sense of Digital Transformation in Education is presented. The anticipated innovation is presented through a school-shared agenda that focuses on technologically driven pedagogical innovation that embraces all spheres of the school's culture. In the ensuing discourse, the components inherent in the innovation process are elaborated.

From the literature analysis [38,39], a case is made for pedagogy as a principal focus in the digital transformation in schools, with technology and knowledge/change management being the drivers. The implication is that the pedagogy hub of the school agenda underpins the schools' digital transformation. Pedagogy offers the school a reason to innovate in the face of changing learning contents or curricula, syllabuses, competencies, and tools for teaching and learning. In their works in the area of pedagogies of multiliteracies, Cope and Kalantzis [60] infer that pedagogy needs to be applied to offer a basis for (i) situated learning experiences in which both in-school and out-of-school experiences intersect to offer skills and knowledge development, (ii) pedagogical approaches that lead learners to conceptualize the learning process and the emerging knowledge from the process, and showcasing this in the internalization of metacognition competence, and (iii) learning that leads to a learner's ability to be a critical thinker, transforming, transferring, and applying knowledge and skills to generate reflective learning competence, and induces the

competence to solve problems. Defining the direction for the innovative pedagogies, AL offers the leverage for the design and implementation of leaner-centred learning scenarios.

Regional, community, and school circumstances differ in interpreting what needs to or could be done; however, we have already made an inference from [27,28,30] that pedagogical approaches in schools need to be driven by 21st-Century skills. This entails (i) learning skills, (ii) literacy skills, and (iii) life skills (see Figure 1). The position of the authors of this current paper is that any pedagogical approach that relegates the learner to the background is doing a disservice to the student and compromising the future survival of the school graduates. AL offers a convenient framework that blends into the DTE agenda in schools.

Practically, in thinking about school pedagogical innovation, various factors come to mind. These include the presence of an innovation policy, the extent of anticipated culture interactive learning, enhancing student learning engagements, and the offering of quality professional teacher–student relationships. Other factors that come to mind are the provision of quality and relevant learning materials, making learning collaborative, authentic, and constructive. With this expectation, the situation lends itself to the use of AL practices in innovative pedagogical designs. To do so, the following AL characteristics are worth consideration by the school actors and stakeholders of the school's innovation. See Figure 3 for Bonwell and Eison's [49] characteristics of AL.

- Students/learners should be active participants in the learning, not mere listeners of the teacher.
- Students/learners should be allowed to make use of their ideas and skills in practical learning spaces, not simply receivers of information.
- Students/learners should be offered opportunities to engage in higher-order thinking activities such as analysing, synthesising, evaluating, and making facts-based decisions.
- Students/learners should be allowed to engage with both peers and the teacher/facilitator.
- Students/learners should be offered the opportunity to make personal inquiries about their learning responsibilities (metacognition), values, attitudes, previous knowledge and experiences, and dispositions.

Taking off from the notion that pedagogy needs to be learner focused and interactive, leads to the question "how could this requirement be met?" The answer is found in the role of technology (digital tools and resources); which provides the affordance for pedagogical innovation to occur. The position of the authors is based on the findings that digital transformation is not about more and more technology in schools; rather, it is about the optimisation of the schools' available digital resources to facilitate the expected change to meet learning outcomes. In this case, the role of technology should be based on the school's digital agenda and address the following:

(i) Facilitating the innovation of the pedagogical process.
(ii) Supporting shifting teaching and learning from traditional to innovative approaches.
(iii) Supporting teacher professional practice and creation of interactive learning scenarios.
(iv) Serving as a medium for knowledge creation.
(v) Access and support the usage of multi-information sources.
(vi) Creating leverage for learners to be co-creators with teachers and peers in the teaching and learning spaces.
(vii) Creating the processes for empowering learners to take ownership of their learning duties through self-directed learning and receiving feedback for their academic progression.

Having the idea of pedagogical innovation and what technology can do is not enough for digital transformation to occur in a school. There is a need for pedagogical innovation ideas and technological roles to be harmonised. From the extracts of the literature, change knowledge/management plays the harmonisation role. Change Knowledge in the model addresses the implementation of the innovation idea and sets in harmony all entities in the school for the change field [38].

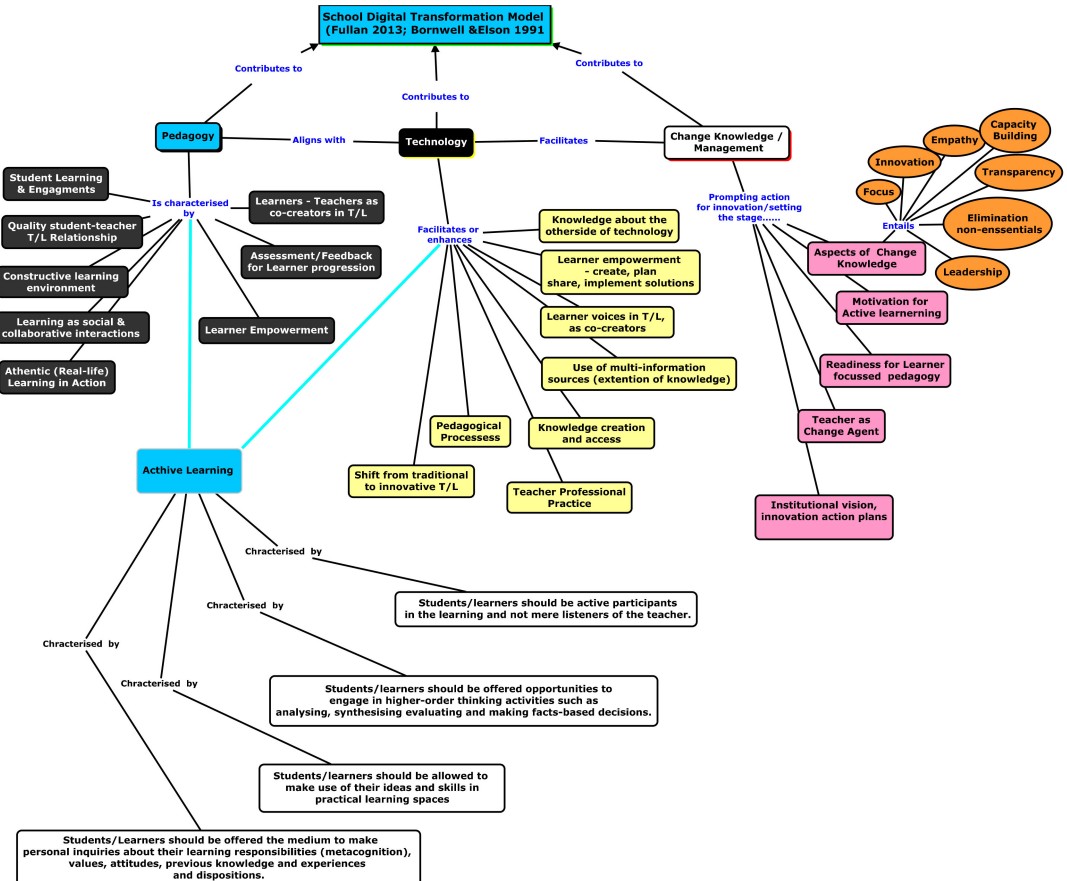

**Figure 3.** Figure designed by the authors with ideas from the works of Fullan's [61] elements for School Innovation representing Digital Transformation in Education Factors, as well as Bonwell and Eison [45], and Chickering and Ehrmaan's Active Learning [49].

The focus of Knowledge Change/knowledge management should be on designing an institutional (school) vision/digital agenda. This vision should spell out clearly what pedagogical innovations the school intends to undertake, and what technologies will be in use. In addition, it should define the procedures, rules, and schedules for the innovation. The policy should focus on teachers as change agents and create the structures to make them co-creators and actors in the planning and decision-making processes, as well as in the implementation of the innovation. Again, the policy should set parameters for the assessment, promote teacher readiness for learner-focused pedagogy, and promote motivation for active learning. As showcased in Figure 3, in addressing the knowledge/management component in Fullan's innovation model, stakeholders should understand that innovation in schools is about "dealing with human beings and not technology"; therefore, values such as empathy, transparency, leadership and building the capacity of people to innovate should be addressed in the schools' digital agenda/policy.

### 3.2. Modelling School-Based Digital Transformation in Education

The overarching base for this current paper is to advocate for school-based management of DTE. Consequently, this section of the paper presents a reference model for planning and implementing school-based managed DTE, as shown in Figure 4. Thus far, we have given the impression that it is essential for DTE in schools to be school managed. The implication is that schools need to own their DTE agenda, with the backdrop that schools' DTE engage in pedagogical innovation, which is facilitated by technology and harmonised through management. As a result, the current paper proposes that it is expedient for school actors to have a reference model from which DTE could be customised for their schools.

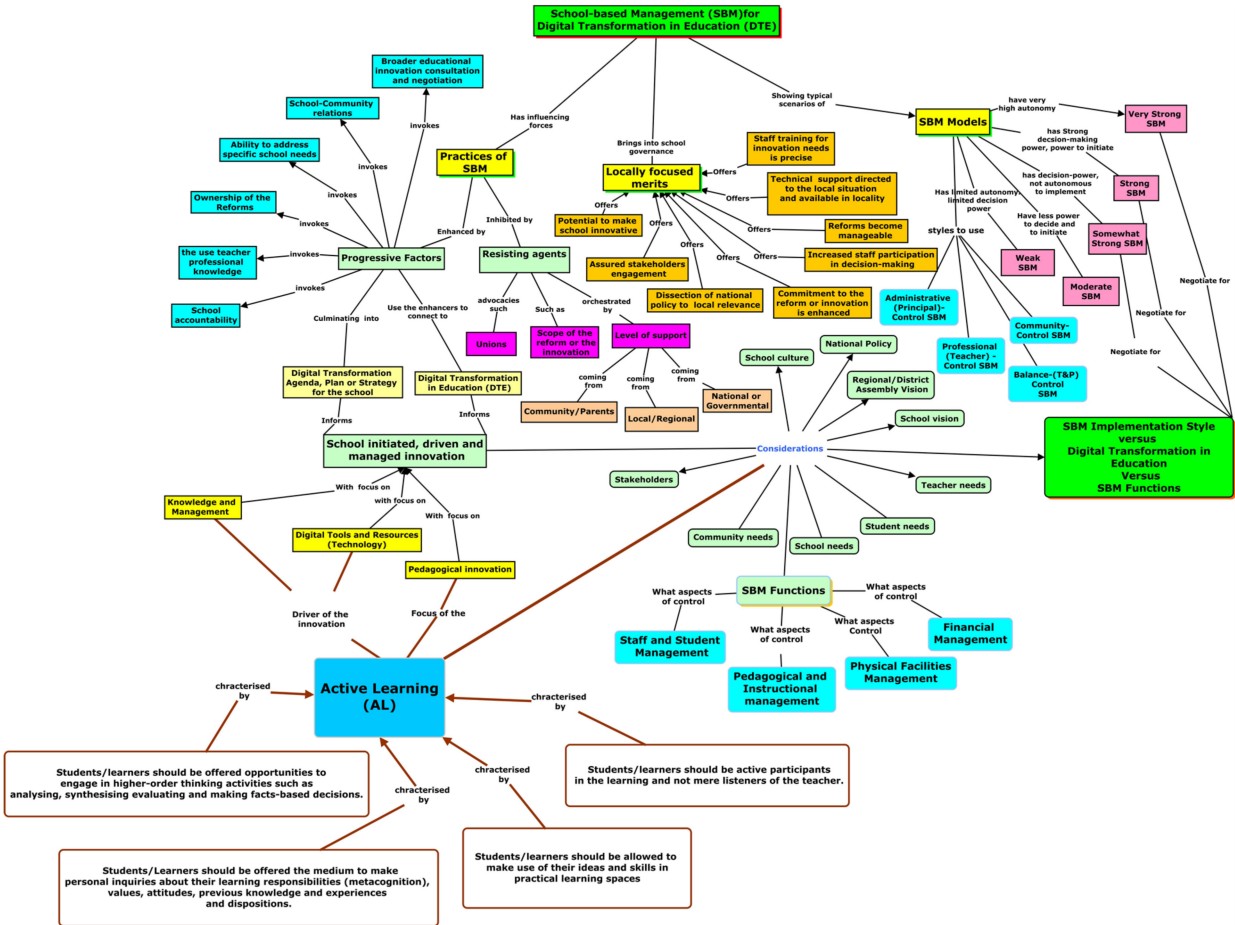

**Figure 4.** A reference model for School-based management of Digital Transformation in Education.

The school-based management of Digital Transformation in Education is the main outcome of the paper and proposes the use of SBM characteristics and principle-run DTE. As seen in Figure 4, an integral scheme of SBM and DTE offers information for designing a school digital agenda. This further informs the contents and focuses of the agenda such that schools are enabled with information to initiate and manage their innovations.

### 3.3. Designing DTE as School-Managed Agenda

Within SBM clusters, various models, practices, merits models, and knowledge management/change indicators are found. School actors should use those to define and visualise the interactivity of the indicators to determine what will go into their DTE agenda or plan. The following is provided in Figure 4:

(a) SBM Model: School actors are pre-disposed to what models of school-based management are applicable in their educational system. These are characterised by how much autonomy could be obtained or at the schools' disposal and could be very strong, strong, somewhat strong, or moderately strong SBM.

(b) SBM Practices: School actors are informed about the positive progress factors and possible resisting factors which should be considered during the design of the DTE plan and management. The possible progress factors include offers of the merits of SBM and providing merits for the pursuit of SBM, and they are indicated by (i) broader educational innovation consultation and negotiations, (ii) enhancing school-community relations, (iii) the ability to address specific school needs, (iv) school actors owning the reform/innovation agenda, (v) using teacher professional knowledge for school innovation, and (vi) enhancing accountability in the school. It is worth noting that these positive indicators do not guarantee the smooth design or implementation of

the SBM innovation agenda, hence the need for the school actors to be conversant with the resisting factors which include (i) teachers/teacher unions, (ii) scope of innovation (unrealistic or ambiguous), and (iii) lack of support from parents, community, local and regional education offices, and/or the government (national policy).

(c)   SBM Merits: In the model (Figure 4), the local (school) stands to gain in using a school-based management approach for its innovation agenda. The evidence of this assertion is characterised by the following indicators: (i) the potential to make school innovation a success, (ii) the assurance of stakeholders' engagement, (iii) the dissection of the national policy to make it relevant at the local level (school), (iv) the possibility of commitment to the reform/innovation, (v) increased staff participation in decision-making, (vi) reforms/innovation becomes manageable, (vii) technical support directed toward the actual local need/situation, and (viii) precise staff training for innovation needs.

(d)   SBM Functions: Like any management or leadership situation, the various functions in SBM are as follows: (i) Staff and student management, (ii) pedagogical and instructional management, (iii) physical facilities management, and (iv) financial management. These are the fundamental frameworks within which the school DTE policy or agenda is situated.

(e)   School-driven/initiated and managed innovation: This cluster of the model taps into the Digital Transformation Plan/strategy to formulate indicators that the school can adopt. These are (i) the pedagogical innovation vision, (ii) available and/or projected digital tools and resources, and (iii) knowledge and change management.

(f)   Active Learning characteristics: In building concepts for interactive, student-based learning (Activate Learning), the model provides information about the characteristics of AL. These are identified as guides for designing AL, namely: (i) in the learning spaces, learners should be active and not passive partakers, (ii) learning should offer learners the opportunity to practice, (iii) activities within the teaching–learning scenarios should elicit higher-order thinking skills in learners, and (iv) learning contents and related trajectories should be personalised for the learner to establish connections.

In principle, the indicators presented in the preceding discourse are meant to support the school actors to carve their local DTE vision(s) and how it is to be managed within the context of SBM. With this backdrop, the ensuing discourse showcases the linkages between the contents and focus of DTE and SBM interplay.

*3.4. Focus on Locally (School) Driven Digital Transformation*

Concerning the indicators supporting the designing of the schools' DTE agenda, this section of the results addresses what could be the contents of the agenda. In Figure 4, the progressive factors of SBM showcase indicators that support the contents DTE should look for. Thus, the school initiated/driven and managed innovation connects directly with DTE, of which the contents of the latter are as follows: (i) pedagogical innovation, (ii) technology (digital tools and resources), and (iii) knowledge/change management.

In the model, the school initiated/driven and managed innovation bubble connects with the integral SBM, DTE, and SBM functions and proposes what should be considered in building the contents of the DTE agenda and management styles to adopt. For what should be considered, the model proposes the following: (a) stakeholder roles, (b) community needs, (c) school needs, (d) student needs, (e) teacher needs, (f) clearly defined school vision/intended school culture, (g) direction of national policy, and (h) direction of regional/direct education offices.

On the question of SBM styles to adopt, the implication is how school actors want the innovation to be controlled. Thus, the options are (i) administrative control, where the principal of the school or school leader controls the process of the DTE design and implementation, (ii) professional control, where teachers' professional know-how is used, or (iii) a combination of both teacher- and school leader-controlled approach to the DTE design and implementation.

The Active Learning bubble plays an overarching role in designing or determining the pedagogical innovation direction and other key considerations and focus of the innovation. AL expands the pedagogical innovation concepts to define what should constitute the content and approach to migrate from passive learning to active one. Thus, by advocating for learner-focused learning innovation leading to the attainment of 21st-Century skills, technology provides affordances for AL to be operationalised. In this process, learning becomes experienced and authentic with support from SBM drivers. Again, AL expands the pedagogical innovation to define what should constitute innovative teaching and learning activities.

In principle, the results presented in this section of the paper are extracted from the model (Figure 4). It offers the actors points and pieces of information to determine which areas would be DTE focus and what management approaches would be adopted. It further suggests an outline of learning scenarios and methods that could be used to support Active Learning engagements and learning activities. In sum, the outcome of this current paper is summarised in Table 1.

**Table 1.** Summary of answers to the guiding questions.

| Leading Questions | Extracted Observations and Inferences from Publications and Theme Clusters (Tracks) in the DTE Reference Model for Stakeholders/School Actors' Conversations |
|---|---|
| 1. What DTE concerns are found in schools and what are the concerns impacting school innovation? | <ul><li>Low use of technology in schools: Wilson and Boateng [12].</li><li>Teacher inhibitors to professional use of ICTs: Agyei [13]</li><li>Lack of stakeholder involvement in school innovation: Ngajie and Ngo [14]</li><li>Schools without ICT vision or strategy: Tondeur et al. [15]</li><li>Utility challenges: Evans and Acosta [2]</li><li>Digital tool availability does not necessarily imply teacher ICT usage: Mukuni [16]</li><li>Lack of leadership for digital innovation and shared vision: Antes and Schuelke [35]</li><li>The need for ICT in schools based on school assessment and intervention: Gondwe [19]</li><li>Lack of shared understanding in schools/management about ICT usage in curriculum implementation: Frans and Pather [43]</li></ul> |
| 2. What could be the possible contents of a DTE framework for a school innovation? | <ul><li>21st-century pedagogies and skills: Partners for 21st-Century Skills [29]</li><li>Teaching actions with digital tools for creativity, critical thinking skills, and problem-solving: Gupta [36]; Balyer and Oz [30]</li><li>Content promoting digital competence readiness: Cambridge Education [27]</li><li>Planning for leaner-centred engagements: Beaudry (2022) [46], Verdiyea [47]</li><li>Training for teacher self-awareness for learner-centred active learning: Sitthiworachart et al. [48]</li><li>Support for teachers to be change agents: Barrera-Osorio et al. [51]</li><li>Training for school-based management for reform: Moradi et al. [33]</li><li>Thinking about bottom-up school invocation actions: Pereira et al. [18]</li><li>Definition of technology roles in the curriculum: Gumede and Baddriparsad [10]</li><li>Innovative pedagogies: Herodotou et al. [31]</li></ul> |
| 3. What could be the focus of a typical locally school driven DTE Agenda? | <ul><li>Pedagogical innovation, Learner-centred pedagogical approach for active learning Technology innovation, Change management, school-based management, and leadership for innovation and policy: Fullan and Watson [39], Miller [38], Verdiyea [47], Veldsman et al. [44], Moradi et al. [33], Elmelegy [55], Fullan ([58], Hargreaves and Fullan [59], V. Hubenakova, D. Sveda, A. Misianikova, and M. Kires [23].</li></ul> |

**Table 1.** *Cont.*

| Leading Questions | Extracted Observations and Inferences from Publications and Theme Clusters (Tracks) in the DTE Reference Model for Stakeholders/School Actors' Conversations |
| --- | --- |
| 4. What DTE broader framework could be proposed to serve as a reference model for school actors to plan their school-based management agenda? | The integral design of DTE reference model recommends various tracks or themes for possible conversations about school-based innovation planning and management.<br>(i) DTE Agenda/Plan, (ii) contents of the plan, (iii) tools and resources, and (iv) leadership/governance. |

## 4. Discussion

The relevance of technology in education cannot be overemphasized; Pereira et al. [18] shared the view that digital transformation in organisations is essential and plays a key role in the viability and sustainability of the organisation. In this paper, a literature review was done on ICT in schools in the SSA region, including best practices, challenges, and ways forward. Using both local and global practices and recommendations, a DTE model for schools is presented. The model integrates SBM and digital information agenda as school driven. Additionally, this is a call for further open conversations about Digital Transformation in Education as a school-based initiative and a school-managed agenda.

The challenge of pursuing DTE in schools comes with very much confusion and erroneous interpretations. The inconsistency emerges from the difference between the definition of ICT in education and how school leaders and teachers interpret the concept. Concerning the literature [24,27], Digital Transformation embraces social, organisational, and technological dimensions. DTE in schools is not only about technology. Though ICT/digital tools play a great part in digital transformation, technology is useless if its value lacks social and organisational context. In this paper, DTE in schools is presented as reflecting the views of Cambridge Education [27], namely (i) the existence of an idea or situation that needs to be enhanced, (ii) a methodology, approach, or process to be improved, and (iii) the deployment of resources (inputs) and harmonising the inputs to improve the situation.

Considering the foregoing, this paper advanced the elements of DTE in schools based on Fullan's ideas, namely pedagogical innovation (change), technology to facilitate the change, and knowledge/change management to harmonise the process. These were then extended using SBM principles and AL indicators to form a complete DTE reference model for schools.

The research acknowledged that one of the challenges of DTE in schools is the lack of DTE vision or agenda (Vey et al. [28]; see Table 1). This lack implies that school actors are without a guide to pursue the DTE agenda. There is a national policy, teachers are given laptops, and computers are deployed in schools, but one typical scenario about how all these could end up at the school level is shared by Mukuni [16], wherein technology in schools did not inform teacher professional development and did not address the digital disparities in schools. It is worth mentioning all around SSA that teachers are informed about the relevance of technology in schools and do have the desire to use technologies in the field (Masingila et al. [40]). The question then is why DTE in schools is still a challenge? As already alluded to, there is no common understanding of DTE in schools amongst stakeholders, as observed by Frans and Pather [43]. We hold the view that for a successful DTE to materialize in schools, schools require a DTE plan that is designed based on the school's unique circumstances. The outcome of this current paper, as the DTE reference model for schools, supports this action with suggested indicators and informing factors.

From the project reference model (Figure 4), the study showcases that a School's DTE agenda cannot be decoupled from the national/regional/district education agenda. The implication is that using all the inputs, including the national policy, school/course curriculum, the school's vision and culture, teachers' needs, students' needs, overall schools' needs, community needs, and stakeholders' roles, a school should be able to set up its

own small-scaled and customised DTE plan. Furthermore, [31,36] assert that 21st-Century pedagogies and skills constitute the essential DTE in schools. Therefore, pedagogical innovations together with all their appendages should be directed at making learners attain the relevant 21st-Century capacities.

Interestingly, research has shown that having a plan alone is not enough; this plan and the vision in it need to be shared, and not all school visions tend to be automatically successful. Fabiana [62] observed that even stakeholders' participation does not necessarily result in school improvement. The implication if stakeholders in the reform do not share the vision, there could be challenges along the path. Research [13,30] suggests that visualised school innovation (vision) should be shared with the actors in the school. We acknowledge that the success of DTE in schools is contingent on planning and sharing the vision with the stakeholders of the school. The results of the study (Figure 4) support the SBM design and management of DTE.

SBM in schools is about empowering school actors to take ownership of the DTE agenda of their schools and to support bottom-up policy design (Pereira et al. [18]). This entails stakeholders' collective design and implementation of the agenda [23,35]. From the projected reference model (Figure 4), DTE in schools could be formulated with consideration of the following cluster concepts:

- From the Progressive Factors: School accountability, use of teacher professional knowledge, ownership of the reforms, ability to address specific school needs, enhanced school–community relations, and broader educational innovation consultation and negotiations.
- From the Locally focused: The potential to make school innovative, active participation of stakeholders assured, national policy dissected to fit the local relevance, commitment to the reform/innovation enhanced, staff involvement in decision-making increased, manageable reform, staff training directed to precise/specific, and technical solutions addressing exact situations.
- From the School initiated(driven) and managed innovation: Connection is made with the factors to consider for school information; this offers multi-linkages to the SBM indicators that could be adopted to manage DTE in the school. In another direction, the school-initiated (driven) innovations connect to the knowledge and change, technology, and pedagogical innovation cluster. Here, the pedagogical innovation is extended to join the Active Learning concept, which in turn offers insights into characteristics to consider when designing innovative teaching and learning activities, namely:
  ○ Students/learners should be active participants in the learning, not mere listeners of the teacher.
  ○ Students/learners should be allowed to make use of their ideas and skills in practical learning spaces, not simply receivers of information.
  ○ Students/learners should be offered opportunities to engage in higher-order thinking activities such as analysing, synthesising, evaluating, and making facts-based decisions.
  ○ Students/learners should be given the opportunity to engage both peers and the teacher/facilitator.
  ○ Students/learners should be offered the opportunity to make personal inquiries about their learning responsibilities (metacognition), values, attitudes, previous knowledge and experiences, and dispositions.

With the results of the study (Figure 4) as the backdrop, we advocate for active school-initiated DTE activities, where schools would focus on the areas of relevance and then ideate for innovation. Again, we advocate that a bottom-up innovation approach should be encouraged when it comes to pursuing DTE in schools [18].

In line with the thoughts of [33,63], designing DTE in schools should be based on facts, research, and well-defined needs and vision. This vision and needs are more visible and clearer at the school level, rather than the national level. Practically, the government might announce the deployment of more computers to schools, but some schools may not need computers; rather, they may need the training to integrate the existing computers they

have for their subject teaching. Hence the need to empower school actors to run their DTE using SBM.

## 5. Conclusions

In the paper, an attempt was made to inspire schools to be proactive in taking ownership of the DTE agenda with a bottom-up policy implementation approach (Pereira et al. [18]). Evidence of national policy in ICT in education has been established [2,7,8,10], and teachers are interested in using digital tools and resources according to Evans and Acosta [2], yet schools maximising the digital tools and resources at their disposal remains a challenge according to Tedla [42], partly owing to lack of school DTE vision/agenda as noted by Pather (2021), and lack of empowerment or over-dependence of what is prevalent in SSA schools' "orders from above" before an action can be taken. Consequently, this paper presents an integrated approach, where DTE is managed at the school level in the framework of SBM [39,45,49–51]. In this way, the national/regional/district ICT in education vision/policy is better customised to have relevance in the school and the school's community. We encourage further conversations on this subject matter in the corridors of policymakers, educationists, school leaders, teachers, students, and all stakeholders in the area pursuing DTE in school. The intent is to support stakeholders to have facts-based and real-world intuition of the existence of divergent schools' needs which does not require global interventions, but rather school situation-specific attention. To this end, the SBM for DTE in the school model is presented as a guide to stakeholders to organise the conversation around DTE, and not as an obligatory order that invokes the success of school innovations.

Like any research activity, this paper has its limitations related to the level of the subjectivity of the report. As already reported, a traditional literature review was adopted as the methodology. Therefore, the comments and inferences made were based on the publications, reports, and articles, and aimed at supporting DTE's innovation conversations for educational practitioners and not exclusively for academicians.

**Author Contributions:** Conceptualization, J.S.Q.; methodology, J.S.Q., A.A.O. and M.L.B.; writing—original draft preparation, J.S.Q.; writing—review and editing, J.S.Q., A.A.O. and M.L.B.; visualization, J.S.Q.; supervision, M.L.B. All authors have read and agreed to the published version of the manuscript.

**Funding:** This research received no external funding.

**Institutional Review Board Statement:** Not applicable.

**Informed Consent Statement:** Not applicable.

**Data Availability Statement:** Not applicable.

**Conflicts of Interest:** The authors declare no conflict of interest.

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
