# Peer review of "School-Based Digital Innovation Challenges and Way Forward Conversations about Digital Transformation in Education"

_education, doi:10.3390/educsci13040344_

Round 1

Reviewer 1 Report

The article is written clearly and in a coherent manner. Most of the article is easy to ready and flows quite well.

Author Response

Query: Is the article adequately referenced?

References reviewed. 

Reviewer 2 Report

The title as well as the introduction raised expectations about your manuscript and research. The topic you are addressing would be a relevant addition to existing literature. Thank you for this valuable contribution. I will structure my feedback in (a) general remarks (these comments cover feedback applicable in the entire manuscript), and (b) specific remarks (feedback on sentence and/or word level). The specific remarks can include a quote from your original manuscript to refer to a specific section. The specific remarks will refer to page (emphasis added in boldface; e.g., 1.15/16) and row(s; e.g., 11.15/16).

General remarks:

The overall manuscript is neat and written concisely—with relevant information for existing literature. One aspect that you can focus on is the overall coherence. The information is there, but it is presented in an illogical order. Second, the punctuation needs improvement. Sometimes semicolon and/or comma’s are missing. I would also like you to pay more attention to consistency of terminology, including the way you write things (e.g., terminology regarding covid-19 [see throughout your manuscript if you have written this in the same way]). I would suggest to have a Native English speaker check your manuscript before submission.

Specific remarks:

p.1.6/7             “was to explore” = There are better ways to describe this (the selection of the verbs is odd).

p.1.10–12        Sentence is redundant.

p.1.13              the literature = a literature.

p.1.15              “various documents” à Such as?

p.1.29/30         Sentence is difficult to read. Restructure it.

p.1.35              An extra space has to be added.

p.1.36/37         I would list the countries in a footnote. It disrupts the reading flow.

p.1                   Terminology I have read on this page: innovation, ICT, digital tools, virtual. At this point I expect that you will clarify this soon. On the second page, you also introduce many new terms such as online, digital, technology, remote, ICT, other digital means, etc. The conceptualization of these are unclear. I do not know if you refer to the same concepts (or that they are slightly different). This has to be clarified (e.g., in a footnote) before you continue your work. Also keep in mind that digital literacy is a different concept as well.

p.1.43              This is an en dash (not an EM dash). Please correct this. This also applies to the remainder of this manuscript.

p.1.38–45        This section reads difficult because each sentence starts with the same. Can you revise this?

p.2.48              The word “in” needs to be replaced by “between”?

p.2.51              The “Secondly” implies there should be a “first”; however, I do not know what that is in your paragraph. Can you clarify?

p.2.57              I think you need to display the reference as: [7 ,8] rather than [7],[8].

p.2.68/69         pre- and post-COVID-19: this has to be clarified what you label as these specific time frames.

p.2.74              “technology usage” = usage needs to be clarified because you can view it is terms of frequency or quality (or both).

p.2.78              “a type of ICT curriculum” à This is oddly phrased.

p.2.80              What do you mean with “in real-life work”?

p.2.85              I do not know what you mean with “multiple impediments”.

p.2.90              Opposite from “orders from above” = bottom-up (you use this term later; you can use it here as well).

p.2.91–98        Spaces, hyphens (these should be em dashes as well). Row 96 also has serious punctuation issues.

p.3.101            The question seems redundant. Can you integrate this with the previous section.

p.3                   This page also contains numerous labels for similar concepts, such as digital divide, digital professional development, learners’ literacy (as compared to digital literacy?), digital services, digital infrastructure, etc. This has to be clarified earlier in your manuscript. You mention all these labels without clarifying them.

p.4.156            When I reach this page, I realize your main concept is digital transformation. This should become apparent earlier in your work. The information on that page is information I need to read sooner to put your study in the current perspective.

p.5.231            What is the question mark doing there?

p.5.239            Space issue in that sentence.

p.6                   The figures you use are unreadable. There is too much information displayed in a small size. Please adjust. The figures need to clarify your work; they should be meaning and complement your take-home message. None of your figures do that because I cannot read them. I would also suggest to place a few figures in your Appendix, especially if you increase their size.

p.7.308–311    Spacing (formatting) issues.

p.7.312            You have a sub header here but you do not display this differently. As a result, I think it is a complete new section. Please adjust.

p.7.312–319    Description is vague. I cannot get a clear image of this model. In particular, the word “regimental” needs clarification. Do you refer to the organization? Or do the location? To the materials? A school is a complex system and referring to it as regimental is too vague.

p.7.338            “very specific” à Avoid using intensifiers (these do not add anything to your message). Go over you work and search for these intensifiers. Improve your argument and these words become redundant.

p.7                   The bullet points belong in the previous section (not in the section where you explain the model).

p.8.364            “response” à Is this the correct word? Do you mean “solution”?

p.8                   What I do not know at this point, is to what education you refer. The school—as a system—is different in primary, secondary and tertiary education. Schools can also differ in their philosophy or—for the lack of a better word—paradigm. The school year is also relevant.

p.8                   The core is every time “good learning practice” and you add conditions to that what is considered good learning. The way you present it now is so confusing. It can be more concise. Please revise.

p.8.374/375     You list the author twice.

p.9.448            What does the “/” mean here?

p.9.453            Capital letter.

p.9.462            En dash needs to be replaced by an em dash (if that is what you want to use here).

p.9.465            “simple traditional” = redundant.

p.9                   I find it difficult to follow your reasoning, especially in row 476 till 478. There is no valid argument to not adopt a scientific sound approach. Moreover, you list “several papers and reports”, but I do not know to what documents you are referring. You list the word “themes” in row 479, but aren’t these the inclusion criteria? You can place them in certain themes or overarching categories. In row 486 you mention sources/databases. Do you mean databases? Because earlier in the section you list a few. I also see Google Scholar as a Search engine. I am wondering why. Google Scholar does not work with peer review. This is a quality check of scientific work. Your work does not meet the quality we expect at this point.

From this point onwards, I stopped making notes because the manuscript lacks scientific rigor and lacks information to replicate the study.

Author Response

Response to queries:

Specific remarks:

p.1.6/7             “was to explore” = There are better ways to describe this (the selection of the verbs is odd).

Change made

p.1.10–12        Sentence is redundant.

Adjustments made

p.1.13              the literature = a literature.

Correction made accordingly.

p.1.15              “various documents” à Such as?

Change made

p.1.29/30         Sentence is difficult to read. Restructure it.

Texts re-organised 

p.1.35              An extra space has to be added.

Adjustment done.

p.1.36/37         I would list the countries in a footnote. It disrupts the reading flow.

We consider this not relevant.

p.1                   Terminology I have read on this page: innovation, ICT, digital tools, virtual. At this point I expect that you will clarify this soon. On the second page, you also introduce many new terms such as online, digital, technology, remote, ICT, other digital means, etc. The conceptualization of these are unclear. I do not know if you refer to the same concepts (or that they are slightly different). This has to be clarified (e.g., in a footnote) before you continue your work. Also keep in mind that digital literacy is a different concept as well.

This is the very reason for this paper so that school practitioners will engage in conversations to evolve meaning from these synonymous terms as relevant to their circumstances.  

p.1.43              This is an en dash (not an EM dash). Please correct this. This also applies to the remainder of this manuscript.

Correction done.

p.1.38–45        This section reads difficult because each sentence starts with the same. Can you revise this?

Sentences restructuring is done accordingly. 

p.2.48              The word “in” needs to be replaced by “between”?

Change done.

p.2.51              The “Secondly” implies there should be a “first”; however, I do not know what that is in your paragraph. Can you clarify?

Clarification made.

p.2.57              I think you need to display the reference as: [7 ,8] rather than [7],[8].

Recommendation adhered to.

p.2.68/69         pre- and post-COVID-19: this has to be clarified what you label as these specific time frames.

This is the most recent occurred educational challenge and does not require any elaborate reporting. Moreso, it is not the focus of the paper.

p.2.74              “technology usage” = usage needs to be clarified because you can view it is terms of frequency or quality (or both).

Clarification made.

p.2.78              “a type of ICT curriculum” à This is oddly phrased.

Text restructured. 

p.2.80              What do you mean with “in real-life work”?

Explanation added in the text

p.2.85              I do not know what you mean with “multiple impediments”.

Explanation given.

p.2.90              Opposite from “orders from above” = bottom-up (you use this term later; you can use it here as well).

Relationship of phrases established.

p.2.91–98        Spaces, hyphens (these should be em dashes as well). Row 96 also has serious punctuation issues.

Restructuring done.

p.3.101            The question seems redundant. Can you integrate this with the previous section.

Corrections made.

p.3                   This page also contains numerous labels for similar concepts, such as digital divide, digital professional development, learners’ literacy (as compared to digital literacy?), digital services, digital infrastructure, etc. This has to be clarified earlier in your manuscript. You mention all these labels without clarifying them.

Basically, we want this part to stay as it is to stimulate the thoughts of school practitioners to have conversations around it. 

p.4.156            When I reach this page, I realize your main concept is digital transformation. This should become apparent earlier in your work. The information on that page is information I need to read sooner to put your study in the current perspective.

Yes, that is the reason why it is placed under the problem section.

p.5.231            What is the question mark doing there?

Deletion made.

p.5.239            Space issue in that sentence.

Cleared.

p.6                   The figures you use are unreadable. There is too much information displayed in a small size. Please adjust. The figures need to clarify your work; they should be meaning and complement your take-home message. None of your figures do that because I cannot read them. I would also suggest to place a few figures in your Appendix, especially if you increase their size.

Font sizes increased.

p.7.308–311    Spacing (formatting) issues.

Technical support may be sought.

p.7.312            You have a sub header here but you do not display this differently. As a result, I think it is a complete new section. Please adjust.

Deletion made.

p.7.312–319    Description is vague. I cannot get a clear image of this model. In particular, the word “regimental” needs clarification. Do you refer to the organization? Or do the location? To the materials? A school is a complex system and referring to it as regimental is too vague.

Texts have been reorganised.

p.7.338            “very specific” à Avoid using intensifiers (these do not add anything to your message). Go over you work and search for these intensifiers. Improve your argument and these words become redundant.

Varying writing styles should not be a problem if the core intended contents are comprehensible enough. 

p.7                   The bullet points belong in the previous section (not in the section where you explain the model).

This point was not clear to us, so no action was taken. We consider this as not a major issue.

p.8.364            “response” à Is this the correct word? Do you mean “solution”?

Clarification made.

p.8                   What I do not know at this point, is to what education you refer. The school—as a system—is different in primary, secondary and tertiary education. Schools can also differ in their philosophy or—for the lack of a better word—paradigm. The school year is also relevant.

Yes, organisations or institutions need to use the inputs of this current paper to engage in conversations about their peculiar circumstances. This is the object of this article.

p.8                   The core is every time “good learning practice” and you add conditions to that what is considered good learning. The way you present it now is so confusing. It can be more concise. Please revise.

Some adjustments were made in the texts.

p.8.374/375     You list the author twice.

Deletion made.

p.9.448            What does the “/” mean here?

Adjustments were made accordingly.

p.9.453            Capital letter.

Corrected.

p.9.462            En dash needs to be replaced by an em dash (if that is what you want to use here).

Addressed accordingly.

p.9.465            “simple traditional” = redundant.

The paper targets practitioners.

Reviewer 3 Report

The paper presents a “simple” traditional literature review to extract best practices and models to showcase authors' position regarding the subject matter under – digital transformation in education.

As authors, 91 papers [primary studies] “was further analysed to cluster them into the context of how they address school innovation issues – from technology and pedagogical perspectives and how they portray school-based management of digital innovations”

The main contribution was “a reference model for planning and implementing school-based managed DT” shown in Fig. 4.

The school-based management of Digital Transformation in Education is a subject that deserves investigation. However, a literature review research needs to be guided by a method that describes how the basic activities of search, selection, quality assessment of primary studies, and synthesis of the results will be carried out.

The paper must structure at least a protocol in the format of a table to consolidate the planning of the literature review, containing: 

  • Research questions: described in the Introduction section.

  • Academic literature sources: “Scopus, Oxford Journals, Springer, Francis and Taylor, Google Scholar, and Science Direct”

  • Gray literature: describe the international institutions' repositories that will be considered in the literature review.

  • Keywords: “school leadership, school management, education, school-based activities, digital teaching, digital innovation in schools, digital transformation, ICT in education, digital teaching and learning and ICT in schools”.

  • Primary studies selection techniques: “simple random and snowballing sampling techniques were adopted in selecting the literature used”.

  • Consider other relevant information to the literature review, such as search strings and publication year boundaries.

Necessary adjusts in the text: 

  • Line 409 “(Malen [...]” → remove the parentheses

  • Line 414 “defined in this article SBM [...]” → remove “SBM”

  • Figure 2 - table that presents “Themes extracted from articles” is unreadable. In addition, it needed to be clarified how the themes were extracted (Figure 2) from the reading and analysis of the primary studies..

It is recommended that a table be inserted to summarize the source of each element of the reference model (Figure 4) created from the result of the literature review.

The discussion section should also describe a roadmap for applying the Reference Model to guide a school DTE journey.

A seção de conclusão deve descrever as respostas obtidas ao final do trabalho para as seguintes perguntas de pesquisa formuladas na introdução:

  • What constitutes Digital Transformation in Education (DTE) in digitally disadvantaged communities?

  • What should be the contents, innovation and method in the DTE framework?

  • And how can DTE be designed and run through a locally(school) managed approach?

  • What could be the focus of locally (school) driven digital transformation? 

  • What kind of school innovation governance or leadership approach could be considered in the quest for Digital Transformation in Education in schools?

The COVID-19 pandemic has accelerated the digital transformation in schools, so a reflection on the topic must be included in the work, considering the primary studies published in the last three years.

Author Response

The paper must structure at least a protocol in the format of a table to consolidate the planning of the literature review, containing: 

  • Research questions: described in the Introduction section.

  • Academic literature sources: “Scopus, Oxford Journals, Springer, Francis and Taylor, Google Scholar, and Science Direct”

  • Gray literature: describe the international institutions' repositories that will be considered in the literature review.

  • Keywords: “school leadership, school management, education, school-based activities, digital teaching, digital innovation in schools, digital transformation, ICT in education, digital teaching and learning and ICT in schools”.

  • Primary studies selection techniques: “simple random and snowballing sampling techniques were adopted in selecting the literature used”.

  • Consider other relevant information to the literature review, such as search strings and publication year boundaries.

 Response to the above questions addressed. RQ restructured. Table created to showcase results relative to RQ.

Necessary adjusts in the text: 

  • Line 409 “(Malen [...]” → remove the parentheses

Action taken

  • Line 414 “defined in this article SBM [...]” → remove “SBM”

Action taken

  • Figure 2 - table that presents “Themes extracted from articles” is unreadable. In addition, it needed to be clarified how the themes were extracted (Figure 2) from the reading and analysis of the primary studies..

 The figure was enlarged.

It is recommended that a table be inserted to summarize the source of each element of the reference model (Figure 4) created from the result of the literature review.

Table added

The discussion section should also describe a roadmap for applying the Reference Model to guide a school DTE journey.

Sequence for model use added to the Table. 

A seção de conclusão deve descrever as respostas obtidas ao final do trabalho para as seguintes perguntas de pesquisa formuladas na introdução:

  • What constitutes Digital Transformation in Education (DTE) in digitally disadvantaged communities?

This is the objective of the article,  institutions or communities will have to define where they belong based on their digital culture or status. We prefer to let this term stands as it is.

  • What should be the contents, innovation and method in the DTE framework?

  • And how can DTE be designed and run through a locally(school) managed approach?

  • What could be the focus of locally (school) driven digital transformation? 

  • What kind of school innovation governance or leadership approach could be considered in the quest for Digital Transformation in Education in schools?

Research questions adjusted

The COVID-19 pandemic has accelerated the digital transformation in schools, so a reflection on the topic must be included in the work, considering the primary studies published in the last three years.

This is not the focus of the article and we prefer it does contain very little focus as already provided in the article.

Round 2

Reviewer 2 Report

I would still ask an English Native speaker to go over your work (the grammar is still erronous). 

Author Response

Query: "I would still ask an English Native speaker to go over your work (the grammar is still erronous)"

Response: Proofreading is done.